# Validation and refinement of existing methods for timed mating and early pregnancy detection in guinea pigs

Marina Luisa Mayer[1], Ilja Finkelberg[2], Elvira Mass[1]*

1 Developmental Biology of the Immune System, Life and Medical Sciences (LIMES) Institute, University of Bonn, Bonn, Germany, 2 Children's Hospital, Pediatrics II, Pediatric Nephrology, University of Duisburg-Essen, Essen, Germany

* elvira@uni-bonn.de

## Abstract

Biomedical research increasingly focuses on early developmental stages to better understand homeostasis, environmental influences, and disease origins, an approach summarised by the concept of the *Developmental Origins of Health and Disease*. However, studying these processes in humans is challenging, highlighting the need for suitable animal models that closely mimic human pregnancy and early development. Historically, guinea pigs were the preferred model for specific research areas and are now regaining attention in embryological studies due to their physiological similarities to humans. We present an extended and standardized protocol for mating guinea pigs, building upon the methods described by Wilson et al. (2021). In this work, we verified the previously published procedures for monitoring the estrous cycle through observation of the vaginal closure membrane and for confirming pregnancy via ultrasound. In addition, we introduced new methods to improve breeding efficiency and early pregnancy assessment. Specifically, we incorporated vaginal cytology alongside vaginal membrane monitoring to increase the success rate of overnight matings. Extending on the ultrasound descriptions by Wilson et al., we provide new ultrasound-based observations of early pregnancy, including the earliest visualization of the embryonic sac at embryonic day 12 (E12) and examples of embryonic development during the first weeks of gestation. By following these step-by-step instructions, researchers new to guinea pig models can quickly establish the methodology in their laboratories, reducing the need for prolonged trial and error. This approach facilitates the broader use of guinea pigs in developmental and reproductive research, particularly due to their resemblance to humans in pregnancy, embryonic development, and parturition.

## Introduction

Guinea pigs (*Cavia porcellus*) have served as an important animal model in biomedical research since the early 19th century, contributing to numerous foundational

**Data availability statement:** All relevant data are within the manuscript and its Supporting Information files.

**Funding:** Funded by the Deutsche Forschungsgemeinschaft (DFG, German Research Foundation) under Germany's Excellence Strategy-EXC2151-390873048 (to EM), European Research Council (ERC) under the European Union's Horizon 2020 research and innovation program (Grant Agreement No. 851257, to EM). This publication was published at the University of Bonn and supported by the Open Access publication fund of the University of Bonn.

**Competing interests:** The authors have declared that no competing interests exist.

discoveries and therapeutic advances. They are particularly valuable in areas where their physiological, immunological, or developmental features more closely resemble those of humans than those of other commonly used rodent models such as mice or rats [1]. Notably, guinea pigs played a central role in pioneering studies on infectious diseases, including bacterial [2], pulmonary [3], and sexually transmitted infections [4]. Owing to their anatomical and neurodevelopmental similarities to humans, especially within the auditory system, guinea pigs have become an important model for auditory research [5]. Their use by researchers such as Robert Koch [6] and Emil von Behring [7] contributed to landmark discoveries that were later recognized with Nobel Prizes. Beyond infectious disease and sensory research, guinea pigs are also increasingly recognized as a powerful model for studying prenatal and postnatal development [8,9]. Since developmental processes cannot be easily investigated in humans, animal models play a critical role in this field. Guinea pigs are particularly well suited for such studies due to key similarities in placental structure, implantation patterns, neuronal development, and pregnancy progression [10,11]. As a result, they have been widely used in studies investigating preeclampsia, placental barrier function, and placental transfer [12–15].

Although guinea pigs have been studied extensively in the context of developmental biology and reproduction, many procedures, such as standardised protocols for successful timed mating, pregnancy confirmation and embryonic development remain limited in the literature. Given the comparatively long estrous cycle (15–17 days) and gestational period in guinea pigs, as well as the narrow ovulation window of only a few hours, precise timing of mating is essential to achieve successful pregnancies [16]. While external signs of receptivity, such as vaginal membrane perforation, can be used to guide mating, these require regular and careful monitoring by trained personnel [17]. For developmental studies, timed mating is particularly critical to reliably stage embryos across gestation. However, existing literature largely describes prolonged cohabitation or mating windows over several days, which impairs precise staging. In this protocol, we describe an approach to increase timed mating efficiency in guinea pigs. Our method builds upon and extends established techniques by Wilson et al. (2021) and others, combining daily assessment of the vaginal membrane with vaginal cytology [16–19]. In contrast to mice, where mating can be confirmed by the presence of a vaginal plug, such identification in guinea pigs is unreliable: plugs may be expelled shortly after mating or not be detectable at all [16,17]. Thus, we provide a decision tree that can be used by unexperienced scientist to achieve efficient timed mating.

Furthermore, gestational weight gain becomes a reliable pregnancy indicator only in mid-gestation. While one study reports that pregnancy can be confirmed via ultrasound as early as embryonic day 16 (E16) [20], more recent studies describe reliable detection from approximately E20 onward [17,21]. To enable early treatment paradigms that can be tested in guinea pigs during the first trimester, we aimed to improve pregnancy detection using various ultrasound devices and successfully detected gestational sacs as early as E12. Additionally, we provide morphological documentation of embryonic development from E15 to E27.

 

## Materials and methods

The protocol described in this peer-reviewed article is published on protocols.io, https://dx.doi.org/10.17504/protocols.io.kqdg31xpql25/v1 and is included for printing as supporting information file 1 with this article.

## Expected results

Guinea pigs offer several advantages as animal models for developmental research; however, researchers unfamiliar with the species may struggle with key procedures for successful mating and pregnancy detection. Unlike mice, in which the estrous cycle can be assessed by simple visual inspection of the external genitalia [22], a widely accepted and practiced method, the same is not straightforward in guinea pigs. In guinea pigs, the vaginal closure membrane, composed of epithelial cells and sealing the vaginal orifice during non-fertile phases [16], serves as a useful marker of estrous stage. However, the membrane is not easily visible without gently parting the labia (Fig 1), a procedure requiring familiarity and training. For untrained personnel, the open phase, when the membrane is absent or perforated, is typically the easiest to recognize. Regular monitoring of the vaginal membrane can improve mating efficiency, yet mis-timed matings may still occur since ovulation can precede the membrane's rupture. As originally described by Stockard and Papanicolaou, vaginal secretions, specifically their presence and color, can offer additional cues for optimal mating timing [16]. These secretions are also detectable as mucus films on vaginal cytology smears.

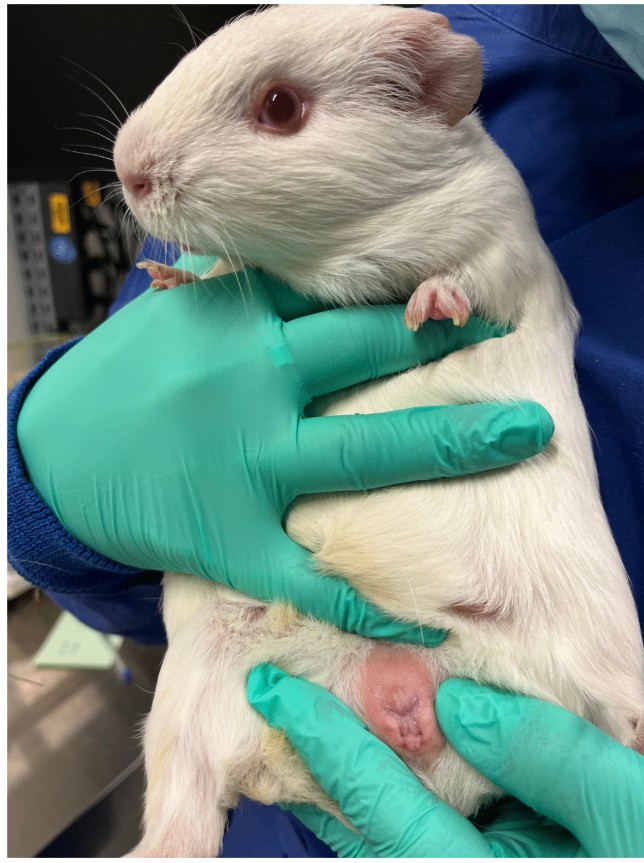

**Fig 1. Examination of the vaginal membrane.**

Vaginal cytology further supports accurate estrous cycle staging. As early as 1917–1919, Stockard and Papanicolaou described four distinct stages based on gradual changes in the vaginal epithelium, with ovulation occurring shortly before leukocyte infiltration of the vaginal wall [16,23]. Copulation is typically restricted to the phase when leukocytes are absent, which is further subdivided into a preparatory phase (when the vaginal lumen appears dry) and a subsequent phase marked by the onset of mucus secretion, corresponding to the „first half" and „second half" during the "closed" stage. Leukocyte migration through the uterine and vaginal epithelium begins in the second phase, corresponding to the "opening" stage, and leukocytes remain prominent during subsequent stages alongside epithelial cells. If fertilization does not occur, early detection of non-pregnancy is essential to make timely use of the subsequent ovulatory period. In mice, detection of a copulatory plug the morning after mating is standard; however, plug formation in guinea pigs remains debated. In our observations, a true copulatory plug was never identified, though seminal residue was occasionally visible on the vaginal opening. It remains unclear whether guinea pigs do not form plugs, whether plugs are expelled rapidly, or whether identification requires more advanced training.

As an alternative, we attempted to detect sperm via PCR targeting the Y-chromosome using previously published primers for *Dystrophin* and *Sry* [24]. These were validated on male and female liver tissue as well as sperm isolated from testicular tissue. Despite optimization via temperature gradient testing and extended cycling protocols, swabs from the vaginal opening post-mating failed to yield sufficient material for sperm detection, even in cases where mating was confirmed by resulting pregnancy. Therefore, the combination of visual inspection of the estrous cycle (Fig 2), vaginal cytology (Fig 3) and ultrasound remains the most reliable method for early pregnancy confirmation. While previous studies reported

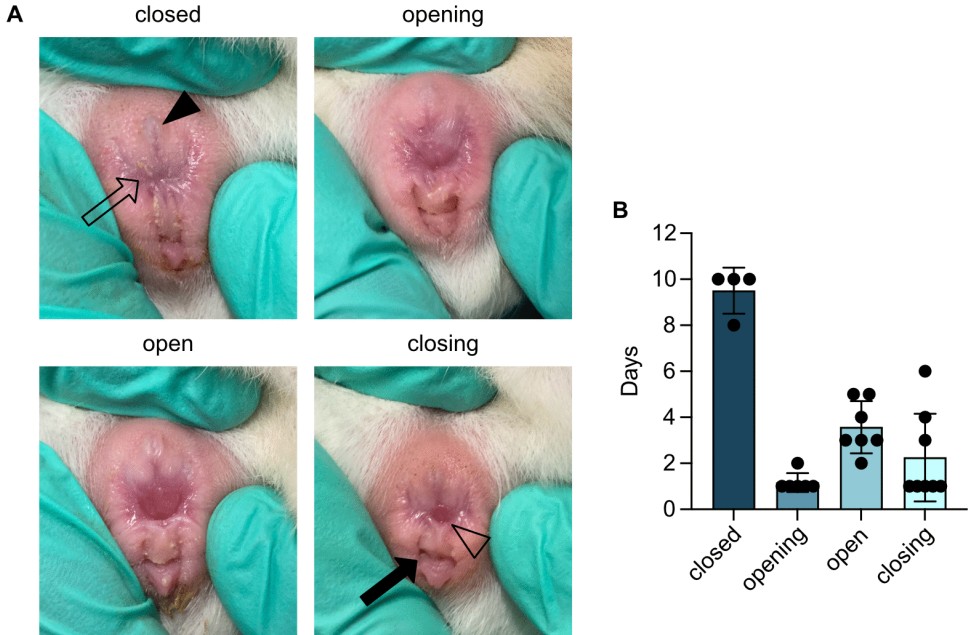

**Fig 2. Changes in the vaginal membrane during different phases of the estrous cycle.** (A) Closed vaginal membrane: The female external genitalia form a visible trident shape when the vaginal membrane is closed. Opening: The vaginal membrane begins to separate, and the clitoris appears swollen and changes colour from pale to a darker pink. Open: The vaginal membrane is fully perforated and the external vaginal opening is visible as a dark pink opening. Closing: The vaginal membrane begins to close from the outer edges, and the swelling subsides. Filled arrowhead: urethral opening, open arrow: vaginal membrane, filled arrow: anus, open arrowhead: vaginal opening. (B) Duration of each phase of the estrous cycle based on changes in the vaginal membrane (each dot represents one animal; mean ± SD).

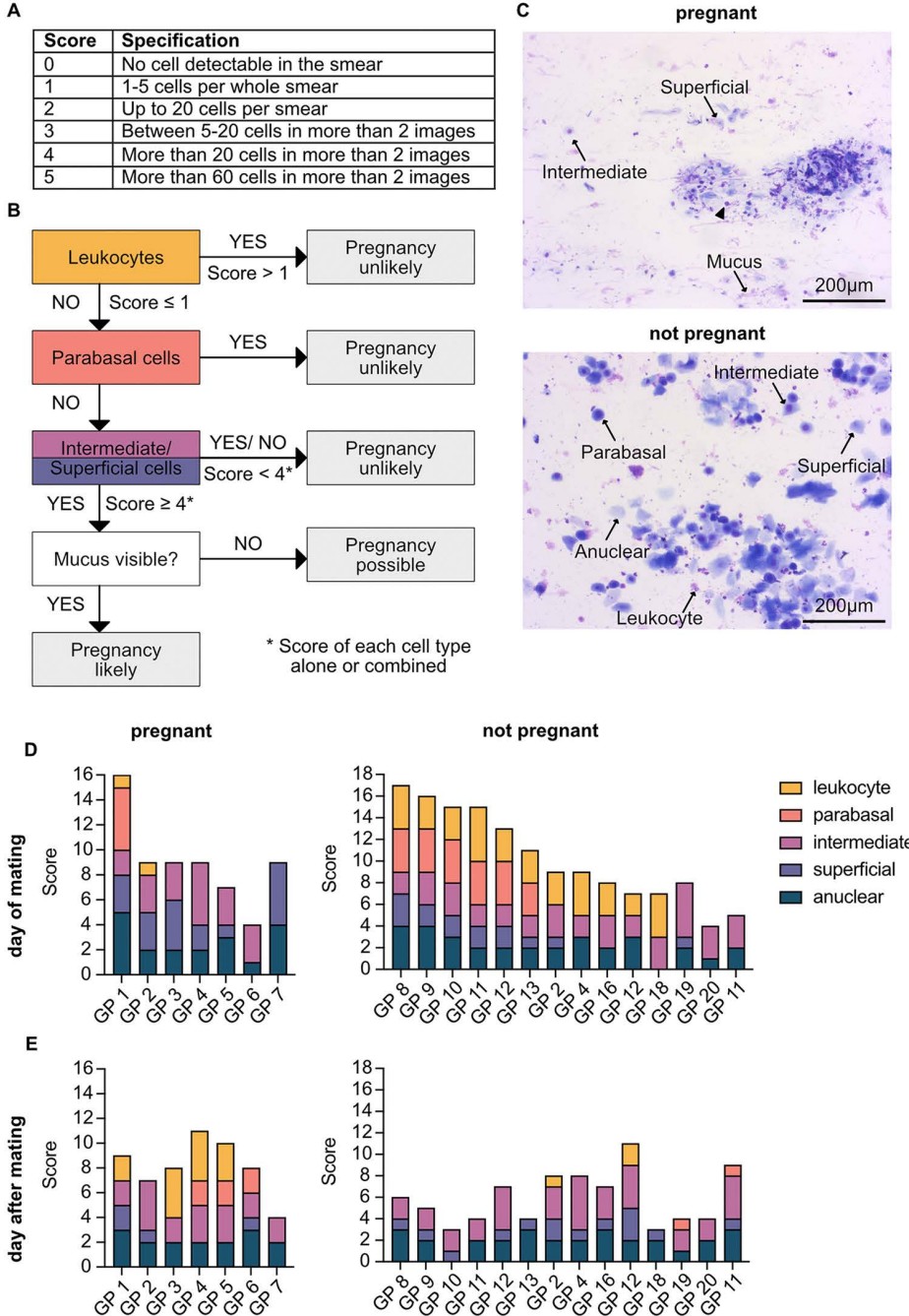

**Fig 3. Vaginal cytology as a measure of fertility.** (A) Scoring system based on the relative presence of different vaginal cell types. (B) Decision-making scheme to assess the likelihood of a positive pregnancy outcome. (C) Representative vaginal smears from the day of mating in one pregnant and one non-pregnant animal. In the pregnant animal, fragmented cells and mucus are visible, whereas in the non-pregnant animal, large, well-defined cells predominate. (D) Distribution of cell types on the day of mating in animals that were later confirmed to be pregnant or failed to conceive. (E) Changes in cell type distribution one day after mating in animals that were later confirmed to be pregnant or failed to conceive.

that ultrasound detection of pregnancy is possible by E16 [20], our findings indicate it is feasible as early as E12 (Fig 4). At this stage, while direct visualization of embryos or placental tissues may be challenging, gestational sacs can be clearly identified due to their thick-walled structure. Visualization of both placental tissue and the embryo is more challenging and may only be achieved by imaging from multiple angles at this early stage. Accurate identification requires scanning the correct anatomical region, as large blood vessels can resemble gestational sacs (S2 File).

By approximately E20, visualization of the embryo and placenta becomes increasingly distinct, eliminating the need to rely solely on gestational sac visualization (S3 File). In later gestation (around E36), the embryo has developed into a fetus with distinguishable anatomical features. At this stage, ultrasound can detect the spine, other skeletal structures, and the fetal heartbeat (S4 File) [20]. During the second half of gestation, as maternal weight gain becomes more pronounced, pregnancy can also be confirmed by cautious abdominal palpation.

For developmental studies, embryos must be carefully dissected from the uterus by incising the uterine wall using blunt-tipped scissors to minimize tissue damage. Embryos are attached to the decidua via the subplacenta, from which they must be gently separated. Each embryo is enclosed within the parietal yolk sac (Fig 5A-D), through which it remains connected to the placenta via the umbilical cord. Isolation of the embryo requires the stepwise removal of multiple tissue layers. The outermost layer is the highly vascularized parietal yolk sac, followed by the visceral yolk sac. The embryo resides within the amniotic cavity, enclosed by the amnion, a thin, nearly transparent membrane. The embryo measures around 3 mm in length by E15. A balloon-like extension from the caudal region of the embryo is apparent (Fig 5E), likely representing the developing allantois. Between E15 and E17, the embryo grows to a size of around 5 mm in length and a

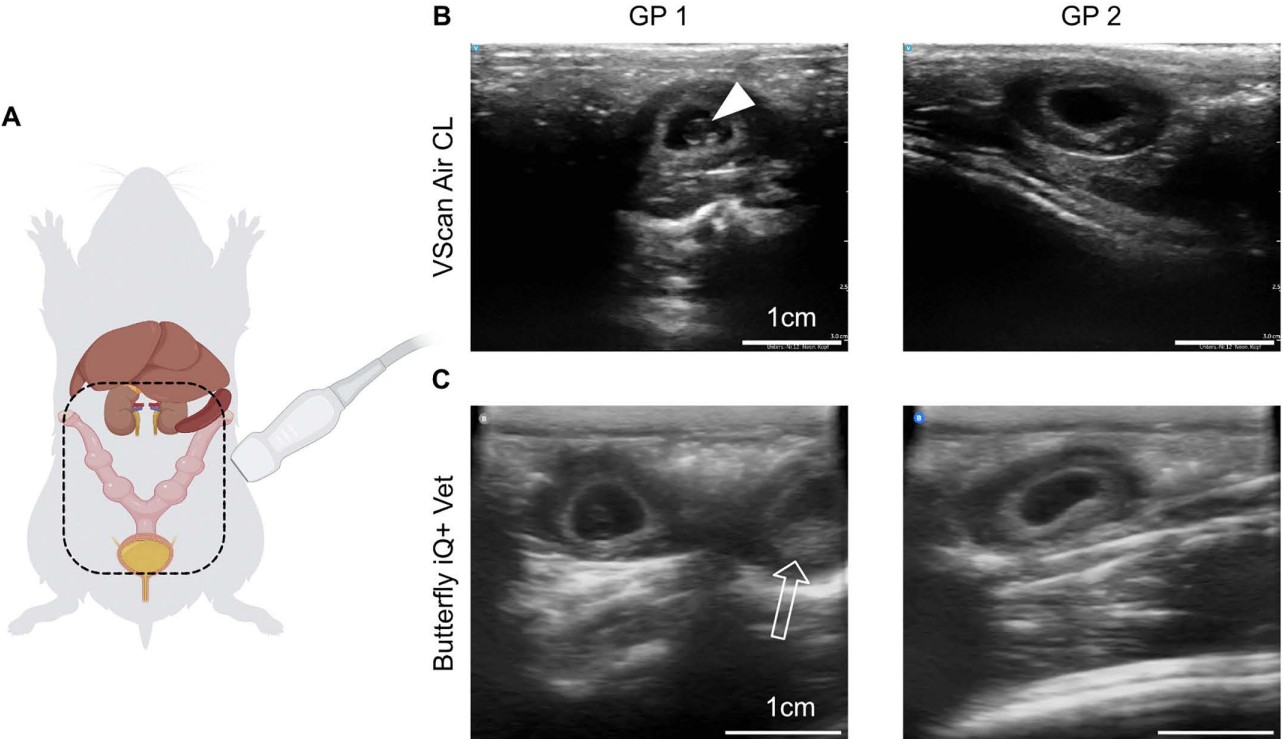

**Fig 4. Ultrasound examination of embryos at E12.** (A) Schematic showing the anatomical position of internal organs and the pregnant uterus. The dashed outline indicates the area that must be shaved prior to ultrasound examination. (B) Ultrasound images from two pregnant animals obtained using the VScan Air CL, the filled arrowhead indicates a visible embryo. (C) Ultrasound images from two pregnant animals obtained using the Butterfly iQ + Vet, the open arrow indicates a visible placenta.

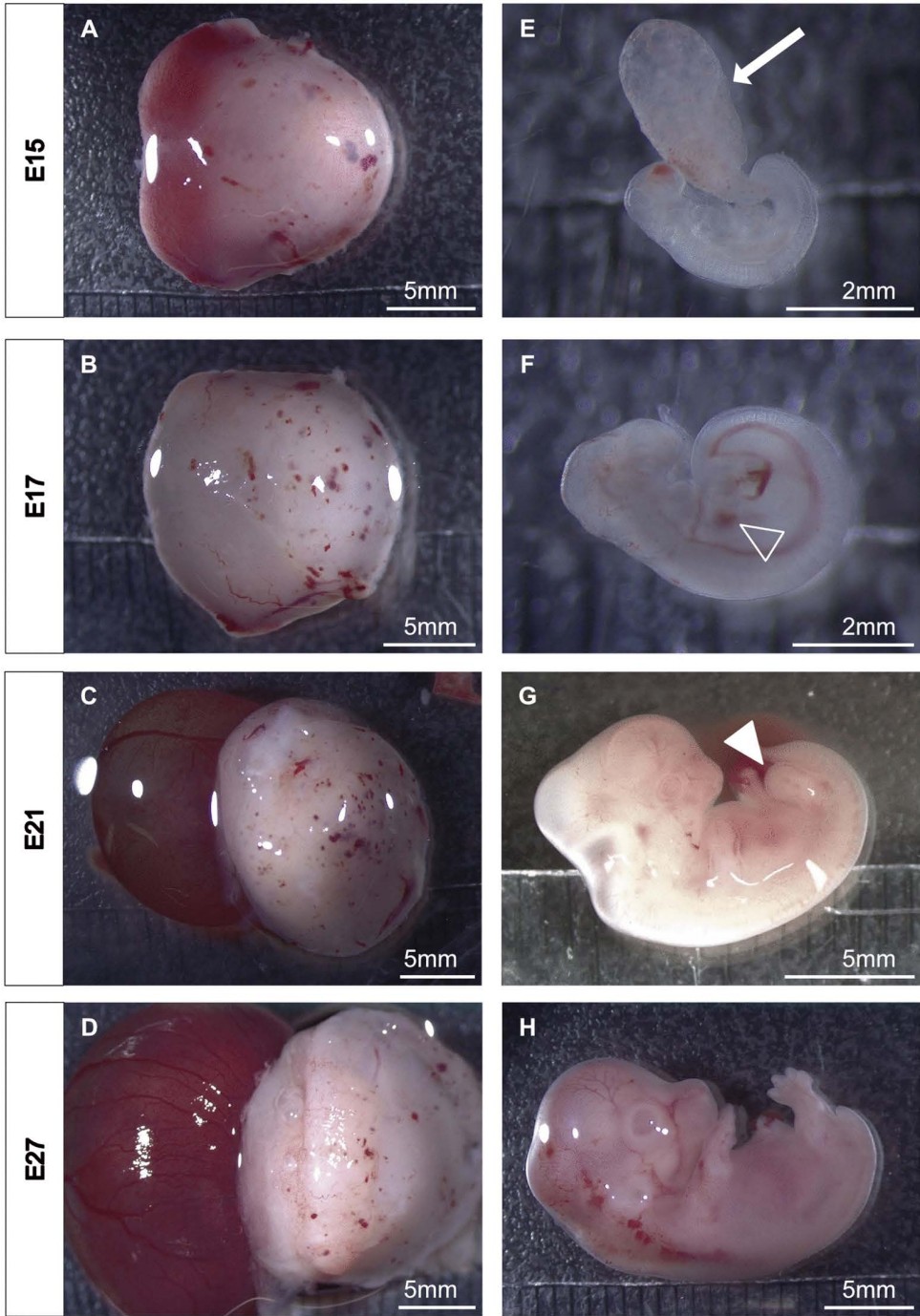

**Fig 5. Morphological documentation of embryonic development from E15 to E27.** (A-D) shows the whole gestational sac with the yolk sac containing the embryo oriented to the left, and the placenta oriented to the right. (E-H) shows the embryos after embryonic layer removal, with the head oriented to the left and the caudal region oriented to the right. (E) Filled arrow pointing to the allantois. (F) Open arrowhead pointing to the heart. (G) Filled arrowhead pointing to the hindlimb bud.

functional circulatory system becomes visible, and heartbeats can be observed (Fig 5F, open arrowhead, S5 File). At E21, the brain vesicles become distinct, and the embryo transitions from a transparent to an opaque appearance and reaches a size of around 13 mm in length (Fig 5G). Limb buds of the forelimbs and hindlimbs are present and begin to take shape (Fig 5G, filled arrowhead). By E27, the embryo reaches a length of 20 mm with a notable increase in head size, developing external ear anlagen and clearly distinguishable eyes forming. The forelimbs and hindlimbs develop digital grooves, and the tail begins to regress (Fig 5H).

In summary, this protocol enables early and efficient timed mating in guinea pigs and facilitates precise identification and dissection of embryos during the first trimester.

## Supporting information

**S1 File. Detailed Protocol.**
(PDF)

**S2 File. Ultrasound examination of two gestational sacs at E12.** Ultrasound recording from a pregnant guinea pig at E12 obtained using the Butterfly iQ+Vet, showing two gestational sacs containing an embryo and placenta.
(MP4)

**S3 File. Ultrasound examination of an embryo at E20.** Ultrasound recording from a pregnant guinea pig at E20 obtained using the VScan Air CL with doppler ultrasound on the embryo to visualise the blood flow in particular in the heart of the embryo.
(MP4)

**S4 File. Ultrasound examination of a fetus at around E36.** Ultrasound recording from a pregnant guinea pig at E36 obtained using the Butterfly iQ+Vet, showing the spine and beating heart of the fetus.
(MP4)

**S5 File. Embryo at E17 with a visible beating heart.** Isolated embryo at E17 where circulation and the heart beat are visible.
(MP4)

## Author contributions

**Conceptualization:** Marina Mayer, Elvira Mass.

**Data curation:** Marina Mayer, Ilja Finkelberg.

**Formal analysis:** Marina Mayer, Ilja Finkelberg.

**Funding acquisition:** Elvira Mass.

**Investigation:** Marina Mayer, Ilja Finkelberg.

**Methodology:** Marina Mayer, Ilja Finkelberg, Elvira Mass.

**Project administration:** Marina Mayer, Elvira Mass.

**Supervision:** Elvira Mass.

**Validation:** Marina Mayer.

**Visualization:** Marina Mayer.

**Writing – original draft:** Marina Mayer, Elvira Mass.

**Writing – review & editing:** Marina Mayer, Ilja Finkelberg, Elvira Mass.

## Acknowledgments

We thank Cornelia Cygon, Dr. Nelli Blank-Stein, Dr. Maria Römelt, Dr. Linda Müller, Dr. Nikola Makdissi, Dr. Katharina Mauel, Dr. Eliana Franco Taveras and Theresa Eulgem for their support with guinea pig handling and Dr. med. Metin Cetiner for his help with ultrasound examination.

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
