## [Decision Letter · Decision Letter 0]

22 Oct 2025

PONE-D-25-45259

Efficient timed mating and early pregnancy detection in guinea pigs

PLOS ONE

Dear Dr. Mass,

Thank you for submitting your manuscript to PLOS ONE. After careful consideration, we have decided that your manuscript does not meet our criteria for publication and must therefore be rejected.

I am sorry that we cannot be more positive on this occasion, but hope that you appreciate the reasons for this decision.

Kind regards,

Kathleen R. Pritchett-Corning, D.V.M.

Academic Editor

PLOS ONE

Additional Editor Comments:

This paper is surprisingly derivative of a cited work, that of Wilson et al. This issue goes beyond normal citation practices and represents a fundamental mischaracterization of the work's originality, as the authors present as novel . The vaginal cytology and qPCR sperm detection protocols are worth publishing. However, this manuscript, as currently written, does not provide sufficient novel contribution to warrant publication as an independent protocol. The manuscript might work if reframed as "Validation and refinement of the Wilson et al. protocol with earlier pregnancy detection and vaginal cytology." The authors likely did not intend to misrepresent their work, but the current presentation could mislead readers about the protocol's origins and the authors' contributions to the field.

Reviewer's Responses to Questions

**Comments to the Author**



Reviewer #1: No

2. Has the protocol been described in sufficient detail?

Reviewer #1: No

3. Does the protocol describe a validated method?

Reviewer #1: No

4. If the manuscript contains new data, have the authors made this data fully available?

Reviewer #1: No

**5. Is the article presented in an intelligible fashion and written in standard English?**

Reviewer #1: Yes

Reviewer #1: The manuscript by Mayer et al. presents a laboratory protocol for timed mating and early pregnancy detection in guinea pigs, aimed at facilitating their use in developmental research. The authors describe methods for estrous cycle monitoring, mating timing, qPCR detection of sperm and ultrasound-based pregnancy detection as early as embryonic day 12. While the protocol addresses practical challenges faced by researchers using guinea pig models, the authors need to address the inadequate acknowledgment of prior work and questionable originality.

This manuscript presents a protocol that is strikingly similar to the previously published work by Wilson et al. (2021, Methods Protoc. 4, 58) - reference 19 in the current manuscript. While Wilson et al. is cited, the authors fail to adequately acknowledge that they are essentially reproducing an existing, well-established protocol rather than developing a novel methodology. Both protocols use identical approaches: vaginal membrane monitoring for estrous cycle tracking, the same mating timing strategy, and ultrasound confirmation of pregnancy. The figures showing vaginal membrane changes are remarkably similar in concept and presentation. The authors present their work as if they independently developed these methodologies ("Here, we present a standardized protocol") rather than clearly stating they are adapting/validating an existing protocol. This raises serious concerns about the originality and added value of the current manuscript. The authors must either: (a) clearly reframe this as a validation/adaptation study of the Wilson et al. protocol with appropriate prominent acknowledgment throughout (remove the vaginal membrane monitoring and time-mating procedures and and simply reference the Wilson et al. 2021), or (b) demonstrate substantial novel contributions that justify publication as an independent protocol.

Additionally, the authors need to clarify the inconsistent pregnancy detection claims via ultrasound. The abstract claims pregnancy detection "as early as embryonic developmental day 12 (E12)" but the methods section mentions this wasn't consistently achieved. The authors state "while direct visualization of embryos or placental tissues may be challenging" at E12, which contradicts the confident claims in the abstract. The reliability and success rate of E12 detection across different operators and equipment also needs clarification.

Minor comment: Line 67 "other studies suggest that reliable detection is not possible before E20 (19,21)." Reference 19 actually states "In our experience, an ultrasound is the only definitive method for determining a pregnancy prior to gestational day 30 (Figure 4), but is limited to not being accurate before gestational day 21". Not that it cannot be done. Reference 21 is a paper from 1986 and ultrasound technology is vastly superior now than back then. Please correct.

.

Reviewer #1: No

- - - - -

---

## [Decision Letter · Decision Letter 1]

31 Mar 2026

Validation and refinement of existing methods for timed mating and early pregnancy detection in guinea pigs

PONE-D-25-45259R1

Dear Dr. Mass,

We’re pleased to inform you that your manuscript has been judged scientifically suitable for publication and will be formally accepted for publication once it meets all outstanding technical requirements.

Kind regards,

Muhammad Zubair

Academic Editor

PLOS One

Additional Editor Comments (optional):

There is no novelty in this manuscript and authors represent mischaracterization of the work's originality, as the authors present as novel

Comments from PLOS Editorial Office: PLOS One is designed to communicate original research and research methods. Novelty is not a criteria for publication on PLOS One.

Reviewers' comments:

Reviewer's Responses to Questions

**Comments to the Author**



Reviewer #1: No

2. Has the protocol been described in sufficient detail?

To answer this question, please click the link to protocols.io in the Materials and Methods section of the manuscript (if a link has been provided) or consult the step-by-step protocol in the Supporting Information files.

Reviewer #1: Yes

3. Does the protocol describe a validated method?

Reviewer #1: Yes

4. If the manuscript contains new data, have the authors made this data fully available?

Reviewer #1: Yes

**5. Is the article presented in an intelligible fashion and written in standard English?**

Reviewer #1: Yes

Reviewer #1: The authors have addressed the comments from the first review. I have nothing further to add.

.

Reviewer #1: No

---

## [Editor Report · Acceptance letter]

PONE-D-25-45259R1

PLOS One

Dear Dr. Mass,

I'm pleased to inform you that your manuscript has been deemed suitable for publication in PLOS One. Congratulations! Your manuscript is now being handed over to our production team.

Kind regards,

on behalf of

Dr. Muhammad Zubair

Academic Editor

PLOS One